# AAD-1: Asymmetric Adversarial Distillation for One-Step Autoregressive Video Generation

**Haobo Li** [1 2] **Yanhong Zeng** [2 3] **Yunhong Lu** [4 2] **Jiapeng Zhu** [2] **Hao Ouyang** [2] **Qiuyu Wang** [2] **Ka Leong Cheng** [2] **Yujun Shen** [2] **Zhipeng Zhang** [1 5]

https://aad-1.github.io/

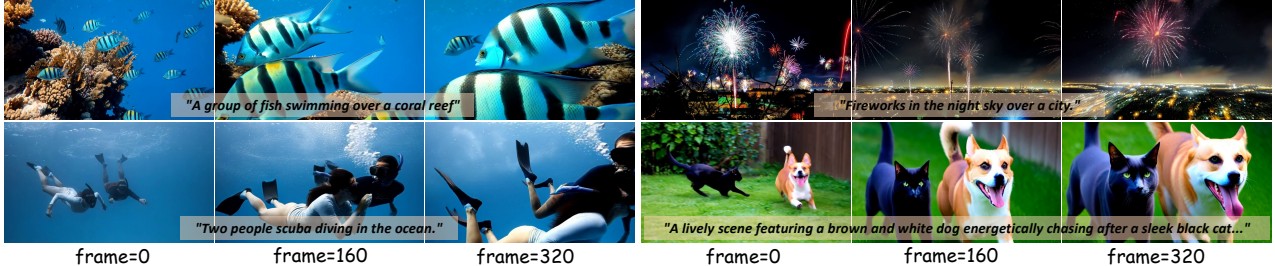

*Figure 1.* We propose **AAD-1**, an **A**symmetric **A**dversarial **D**istillation framework for **One-step** autoregressive video generation. Given a single conditioning image, AAD-1 generates videos autoregressively while maintaining both high visual quality and motion fidelity over long horizons, requiring only one sampling step per chunk.

## Abstract

We present **AAD-1**, an **A**symmetric **A**dversarial **D**istillation framework for **O**ne-step autoregressive image-to-video generation. State-of-the-art methods adopt adversarial distillation but suffer from motion collapse and training instability, resulting in static videos. AAD-1 addresses these challenges through two key designs in architecture and training strategy. Our key architectural insight is to break the symmetry between generator and discriminator. While the generator remains causal to preserve autoregressive sampling capability, the discriminator attends bidirectionally over the full spatiotemporal context and produces a single holistic realism score for the entire video sequence. This asymmetric design enables the discriminator to effectively detect global temporal failures and long-range drift that cause motion collapse in autoregressive generation. To stabilize training, we introduce a phased strategy that first uses distribution matching to bootstrap a stable one-step generator, providing a warm-up phase that brings the student distribution closer to the teacher before adversarial distillation begins. Extensive experiments on VBench demonstrate that AAD-1 achieves state-of-the-art performance in one-step autoregressive video generation.

## 1. Introduction

Fast autoregressive video diffusion post-training has emerged as a promising paradigm that adapts pretrained bidirectional video diffusion models (Wan et al., 2025; Kong et al., 2024; Lin et al., 2024), which are limited to generating fixed-length short clips, into few-step autoregressive models that support indefinitely long video generation (Teng et al., 2025; Yuan et al., 2025). This paradigm has attracted significant research interest due to its value for real-time streaming applications (e.g., gaming) and world modeling (Brooks et al., 2024; Ball et al., 2025; Feng et al., 2024).

Training fast autoregressive video diffusion models presents substantial challenges (Xing et al., 2024; Yin et al., 2025). Recent state-of-the-art methods integrate self-rollout training, where models learn from their own generated trajectories (Huang et al., 2025) rather than ground-truth contexts,

[1]AutoLab, SAI, SJTU [2]Ant Group [3]Department of Automation, Tsinghua University [4]Zhejiang University [5]Anyverse Dynamics. Correspondence to: Zhipeng Zhang <zhipeng.zhang.cv@outlook.com>, Yanhong Zeng <zengyh1900@gmail.com>.

*Proceedings of the 43rd International Conference on Machine Learning*, Seoul, South Korea. PMLR 306, 2026. Copyright 2026 by the author(s).

overcoming the exposure bias in Teacher Forcing (Ho et al., 2022) or Diffusion Forcing (Chen et al., 2024). However, self-rollout training requires performing causal adaptation and step distillation simultaneously, imposing the burden of learning both autoregressive dynamics and accelerated sampling concurrently. This coupled optimization proves particularly challenging, with existing approaches requiring four or more sampling steps to maintain acceptable quality.

In this work, we target the extremely challenging one-step autoregressive image-to-video generation. While adversarial distillation is a leading approach for one-step distillation (Lin et al., 2025a), two critical challenges limit current methods. **(1) Architectural limitation.** Existing methods adopt symmetric discriminator architectures that mirror the generator's causal structure with frame-wise discrimination, as shown in Figure 2-(a) (Lin et al., 2025b). However, a causal discriminator evaluating frame $t$ can only attend to contexts up to block $t - 1$ without future information, causing inherent insensitivity to accumulated temporal degradation. While individual frames appear realistic when conditioned on preceding frames, the overall sequence gradually loses motion fidelity, leading to motion collapse where videos become stuck at the initial frame (Lin et al., 2025b). Aggregating all tokens for a video-level logit (Figure 2-(b)) offers partial improvement, yet causal attention fundamentally limits capturing long-range dependencies. **(2) Training instability.** When training from scratch, early one-step predictions lie far from the data distribution, and under self-rollout training, this gap compounds across time, destabilizing training dynamics (Cheng et al., 2025).

To address these challenges, we propose **AAD-1**, an **A**symmetric **A**dversarial **D**istillation framework for **One**-Step autoregressive video generation with two key innovations in architecture and training. **(1) Bidirectional discriminator with holistic discrimination.** To overcome the architectural limitation, we employ a bidirectional discriminator with video-level holistic discrimination. While the generator remains causal to preserve autoregressive sampling, as shown in Figure 2-(c), the discriminator attends bidirectionally over the full spatiotemporal volume and produces a single realism score for the entire sequence. This asymmetric design provides two critical advantages: (a) the discriminator can detect global temporal failures such as motion collapse that manifest gradually across the sequence, and (b) it can penalize long-range drift by comparing any frame against both past and future context. Our extensive ablations demonstrate that both components are essential, removing either bidirectional attention or video-level scoring substantially degrades motion quality, with causal or frame-wise variants reverting to motion collapse behaviors. **(2) Phased training with distribution matching warm-up.** To stabilize adversarial distillation, we introduce a warm-up stage that leverages frame-wise distribution matching.

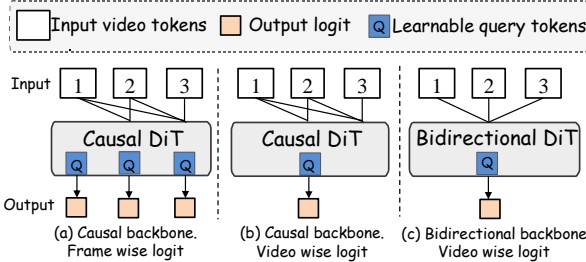

*Figure 2.* **Discriminator Architecture Comparison.** We compare three configurations: (a) **Causal backbone with frame-wise logits**, providing dense local feedback but lacking global temporal context; (b) **Causal backbone with video-level logit**, aggregating information causally but still constrained by unidirectional attention; and (c) **Bidirectional backbone with video-level logit (AAD-1)**, which attends to the full spatiotemporal context. The bidirectional attention in (c) enables holistic discrimination that can detect gradual motion degradation and long-range drift across the entire sequence, which causal architectures are hard to capture.

Specifically, we use DMD to bootstrap a stable one-step generator that produces on-manifold predictions, establishing a foundation for subsequent adversarial refinement. This warm-up phase provides the adversarial stage with initial predictions sufficiently close to real data that the discriminator can provide meaningful gradients, preventing the training instability observed when optimizing from scratch.

We conduct extensive experiments on VBench, demonstrating that AAD-1 achieves state-of-the-art performance in one-step autoregressive video generation with superior visual quality and motion fidelity. Our contributions are:

- We identify critical architectural and training limitations in existing one-step autoregressive video generation that lead to motion collapse and training instability.
- We propose an asymmetric adversarial distillation framework featuring a bidirectional discriminator with video-level holistic discrimination and a phased training strategy with distribution matching warm-up.
- We achieve state-of-the-art one-step autoregressive video generation on VBench.

## 2. Related Work

**Autoregressive video diffusion models.** Autoregressive video diffusion models generate video sequences frame-by-frame, where each frame is synthesized through a diffusion process conditioned on preceding frames (Chen et al., 2025; Zhang & Agrawala, 2025; Wan et al., 2025). Standard training strategies include Teacher Forcing (TF) (Wan et al., 2025; Ho et al., 2022), which conditions on clean historical frames with shared noise schedules, and Diffusion Forcing (DF) (Chen et al., 2024; 2025; Teng et al., 2025), which uses independently noised contexts. To enable efficient streaming inference, recent methods adapt pretrained bidirectional

models by introducing block-causal attention patterns (Yin et al., 2025; Lin et al., 2025b). These patterns apply bidirectional self-attention within local temporal windows while maintaining causal dependencies across blocks, thereby supporting KV-cache reuse during sequential generation.

To further address the train-test distribution gap, several approaches integrate self-rollout training (also termed Self Forcing (Huang et al., 2025) or Student Forcing (Lin et al., 2025b)), where models learn from their own generated trajectories rather than solely from ground-truth data (Liu et al., 2025; Cui et al., 2025). These methods typically perform distillation simultaneously, requiring the model to learn both autoregressive dynamics and accelerated sampling concurrently (Lu et al., 2025b; Hong et al., 2025; Yin et al., 2024b). However, this joint optimization presents significant training challenges, with existing approaches typically requiring four or more sampling steps to maintain acceptable quality (Yang et al., 2025). In contrast, our work targets single-step autoregressive video generation, achieving robust streaming generation with minimal inference cost.

**Accelerating video diffusion models.** Diffusion distillation aims to compress multi-step sampling processes into fewer iterations while preserving generation quality. Existing approaches can be categorized into trajectory-level and distribution-level methods. Trajectory-level techniques approximate the sampling trajectories of teacher models through progressive distillation that iteratively halves the number of steps (Salimans & Ho, 2022), consistency models that map arbitrary trajectory points to their origins (Song et al., 2023), or rectified flow methods that straighten sampling paths (Liu et al., 2022). Distribution-level methods, by contrast, directly match the output distributions between student and teacher models. Representative approaches include adversarial distillation, which employs discriminators to align the distributions of real and generated data (Lin et al., 2025a; Xu et al., 2024; Sauer et al., 2024), and score distillation methods that minimize the reverse KL divergence using the score functions of real and fake distributions (Wang et al., 2023; Yin et al., 2024b;a; Lu et al., 2025a).

In the video domain, existing work largely adapts image distillation techniques to bidirectional models that generate short clips of fixed duration (Shao et al., 2025; Cheng et al., 2025; Mao et al., 2025). APT2 represents the most relevant prior work, applying adversarial distillation to autoregressive video generation (Lin et al., 2025b). Our work differs from APT2 in four aspects. First, APT2 relies on a closed-source model, whereas our study is built on the publicly available Wan 2.1 backbone (Wan et al., 2025) and reports key implementation details of the training recipe. Second, APT2 uses a causal discriminator with frame-wise discrimination; in contrast, we use a bidirectional discriminator with a video-level logit, so the training-time critic

can evaluate a complete rollout with future context. Third, we explicitly separate one-step initialization and adversarial refinement through a DMD warm-up stage, which avoids the instability of cold-start adversarial training. Fourth, we provide controlled ablations of backbone visibility and logit granularity, showing that the causal/frame-wise design is prone to static-video collapse while bidirectional video-level discrimination gives more stable long-horizon generation.

## 3. Preliminaries

**Video notation and sliding-window causal streaming.** We denote a video clip by $x_{1:T} = (x_1, \ldots, x_T)$, where each frame $x_t \in \mathbb{R}^{H \times W \times C}$ (height, width, channels). Let $c$ denote optional conditioning (e.g., text). In sliding-window causal streaming, frames are generated one at a time. At step $t$ the model produces frame $\hat{x}_t$ conditioned on (i) the previous $L$ frames (sliding window of size $L$) and (ii) a set of $S$ *sink frames* $x_{1:S}$ that are always retained from the beginning of the sequence:

$$\hat{x}_t \sim p_\theta\big(\cdot \mid x_{1:S}, \hat{x}_{t-L:t-1}, c\big). \tag{1}$$

The sink frames provide a fixed anchor to the start of the video and help maintain long-range consistency, while the sliding window captures recent temporal context. We write $x_{\text{ctx},t} = (x_{1:S}, \hat{x}_{t-L:t-1})$ to denote the visual context (and omit the subscript $t$ when it is clear). The window slides forward after each frame is generated. Despite these mechanisms, errors can still compound over long sequences, leading to temporal drift.

**Distribution matching distillation (DMD).** DMD transfers a strong teacher diffusion model $p_{\text{T}}$ to a fast student causal generator $G_\theta$ by minimizing a distribution-level divergence. Given noise $z_t \sim \mathcal{N}(0, I)$, visual context $x_{\text{ctx},t}$, and text conditioning $c$, the student produces $\hat{x}_t = G_\theta(z_t, x_{\text{ctx},t}, c)$. The DMD objective encourages $p_{G_\theta} \approx p_{\text{T}}$ using score-based distribution-matching gradients derived from real and fake score estimates. DMD is stable for few-step distillation, but quality can degrade when pushed to a single step.

**Adversarial distillation.** GAN-based distillation trains a causal generator $G_\theta$ together with a discriminator $D_\psi$ that distinguishes real frames from generated ones. The standard adversarial objective is

$$\min_{G_\theta} \max_{D_\psi} \mathbb{E}_x\big[\log D_\psi(x)\big] \\ + \mathbb{E}_{z_t}\big[\log\big(1 - D_\psi(G_\theta(z_t, x_{\text{ctx},t}, c))\big)\big]. \tag{2}$$

For causal streaming, the generator $G_\theta$ must remain strictly causal, producing each frame $\hat{x}_t = G_\theta(z_t, x_{\text{ctx},t}, c)$ using the visual context defined above. The discriminator $D_\psi$

can be either causal (past-only) or bidirectional (accessing future frames during training). This paper studies how discriminator design and a three-stage asymmetric adversarial distillation recipe affect one-step causal generation quality.

## 4. Asymmetric Adversarial Distillation

We study one-step autoregressive image-to-video generation for streaming video applications. Our training pipeline has three stages: (i) ODE initialization via Diffusion Forcing on teacher denoising trajectories under noisy context, (ii) one-step DMD warmup under self-rollout context by matching real and fake scores, and (iii) asymmetric adversarial refinement with a causal generator trained against a bidirectional discriminator with video-level discrimination, see Figure 3.

**Causal architecture adaptation.** We follow the notation in Section 3. In particular, the student causal generator produces one chunk in a single forward pass, $\hat{x}_t = G_\theta(z_t, x_{\text{ctx},t}, c)$, and is deployed autoregressively with a sliding-window visual context $x_{\text{ctx},t} = (x_{1:S}, \hat{x}_{t-L:t-1})$.

**Stage I: ODE initialization.** Following prior work on causal video generation (Yin et al., 2025; Huang et al., 2025), we first use a bidirectional teacher (Wan 2.1 T2V (Wan et al., 2025)) to generate denoising trajectories as supervision targets. We then train the causal student generator $G_\theta$ to regress these teacher trajectories. To align with the few-step inference target (e.g., 1 or 2 steps), we restrict the regression supervision to those specific discrete timesteps used in the downstream stages, rather than the full ODE trajectory. This is implemented via a Diffusion Forcing (Chen et al., 2024) objective where context chunks are noised at levels corresponding to this discrete schedule. Let $\tilde{x}_{\text{ctx},t}$ denote the noisy context and $\mathcal{S}_\phi^{\text{ODE}}(\cdot)$ the ODE-based teacher sampler, the optimization function is defined as:

$$\mathcal{L}_{\text{ODE}}(\theta) = \mathbb{E}_{t, z_t}\left[\left\|G_\theta(z_t, \tilde{x}_{\text{ctx},t}, c) - \mathcal{S}_\phi^{\text{ODE}}(z_t, \tilde{x}_{\text{ctx},t}, c)\right\|_2^2\right].$$

(3)

Autoregressive video generation requires adapting pre-trained bidirectional video models into autoregressive generators by replacing bidirectional full-attention with block-wise causal attention. This stage provides stable initialization for subsequent one-step distillation.

**Stage II: distribution matching warmup.** We employ Self-Forcing Distribution Matching Distillation (Huang et al., 2025) to holistically align the student's autoregressive distribution $p_\theta$ with the teacher's distribution. This framework utilizes three models: the causal student $G_\theta$, a frozen bidirectional teacher $s_{\text{real}}$ (Real Score), and a dynamically updated bidirectional model $s_{\text{fake}}$ (Fake Score). During training, we first perform autoregressive self-rollout to generate a full clip $\hat{x}_{1:T}$ from the student $p_\theta$ using self-rollout context

$\hat{x}_{\text{ctx},t} = (x_{1:S}, \hat{x}_{t-L:t-1})$:

$$\hat{x}_t = G_\theta(z_t, \hat{x}_{\text{ctx},t}, c), \quad t = 1, \ldots, T.$$

(4)

To match distributions, we perturb the entire generated sequence to a random noise level $\tau$ to obtain $\hat{x}_{1:T,\tau}$. The Fake Score model $s_{\text{fake}}$ is trained to estimate the score of the generated distribution via denoising score matching:

$$\mathcal{L}_{\text{score}}(\phi) = \mathbb{E}_{\hat{x}\sim p_\theta, \tau, \epsilon}\left[\|s_{\text{fake}}(\hat{x}_{1:T,\tau}, \tau, c) - \epsilon\|_2^2\right].$$

(5)

Concurrently, the generator $G_\theta$ is updated to minimize the distribution divergence using the gradients derived from the discrepancy between real and fake scores:

$$\nabla_\theta \mathcal{L}_{\text{DMD}} = -\mathbb{E}_{\hat{x}\sim p_\theta, \tau}\left[\left(s_{\text{real}}(\hat{x}_{1:T,\tau}, \tau, c)\right.\right.$$
$$\left.\left. - s_{\text{fake}}(\hat{x}_{1:T,\tau}, \tau, c)\right)^\top \nabla_\theta \hat{x}_{1:T}\right].$$

(6)

Compared to teacher forcing distillation, this self-rollout distribution matching effectively bridges the train-test gap.

**Stage III: asymmetric adversarial refinement.** We refine the one-step generator with adversarial training. We construct a discriminator $D_\psi$ using the Wan 2.1 T2V (Wan et al., 2025) backbone initialized from pre-trained weights. Following the APT (Lin et al., 2025a) architecture, we insert cross-attention heads at the 19th, 29th, and 39th transformer layers to aggregate spatiotemporal features into a scalar score. Unlike APT which operates on clean inputs, we apply Gaussian noise to the discriminator inputs according to a randomly sampled timestep $\tau$. This noise injection is essential for stabilizing the training of our asymmetric generator-discriminator pair. We sample a generated clip $\hat{x}_{1:T}$ by rolling out the causal generator autoregressively:

$$\hat{x}_t = G_\theta(z_t, (x_{1:S}, \hat{x}_{t-L:t-1}), c), \quad t = 1, \ldots, T.$$

(7)

We train a discriminator $D_\psi$ on full clips (hence bidirectional during training), while keeping $G_\theta$ strictly causal. Let $x_{1:T,\tau} = \alpha_\tau x_{1:T} + \sigma_\tau \epsilon$ and $\hat{x}_{1:T,\tau} = \alpha_\tau \hat{x}_{1:T} + \sigma_\tau \epsilon$, with $\epsilon \sim \mathcal{N}(0, I)$, denote real and generated clips perturbed at timestep $\tau$, which is also provided to the discriminator. Using the standard logistic GAN objective, we optimize

$$\mathcal{L}_D(\psi) = -\mathbb{E}_{x\sim p_{\text{data}}, \tau}\left[\log D_\psi(x_{1:T,\tau}, \tau, c)\right]$$
$$- \mathbb{E}_{\hat{x}\sim p_\theta, \tau}\left[\log(1 - D_\psi(\hat{x}_{1:T,\tau}, \tau, c))\right],$$

(8)

$$\mathcal{L}_G(\theta) = -\mathbb{E}_{\hat{x}\sim p_\theta, \tau}\left[\log D_\psi(\hat{x}_{1:T,\tau}, \tau, c)\right].$$

(9)

To stabilize training, we employ approximated R1 and R2 regularizations (Lin et al., 2025a), penalizing the discriminator's sensitivity to small perturbations on real and generated samples, respectively:

$$\mathcal{L}_{\text{reg}}(\psi) = \mathbb{E}_{x,\tau}\left[\|D_\psi(x_{1:T,\tau}, \tau, c) - D_\psi(x_{1:T,\tau} + \delta, \tau, c)\|^2\right]$$
$$+ \mathbb{E}_{\hat{x},\tau}\left[\|D_\psi(\hat{x}_{1:T,\tau}, \tau, c) - D_\psi(\hat{x}_{1:T,\tau} + \delta, \tau, c)\|^2\right],$$

(10)

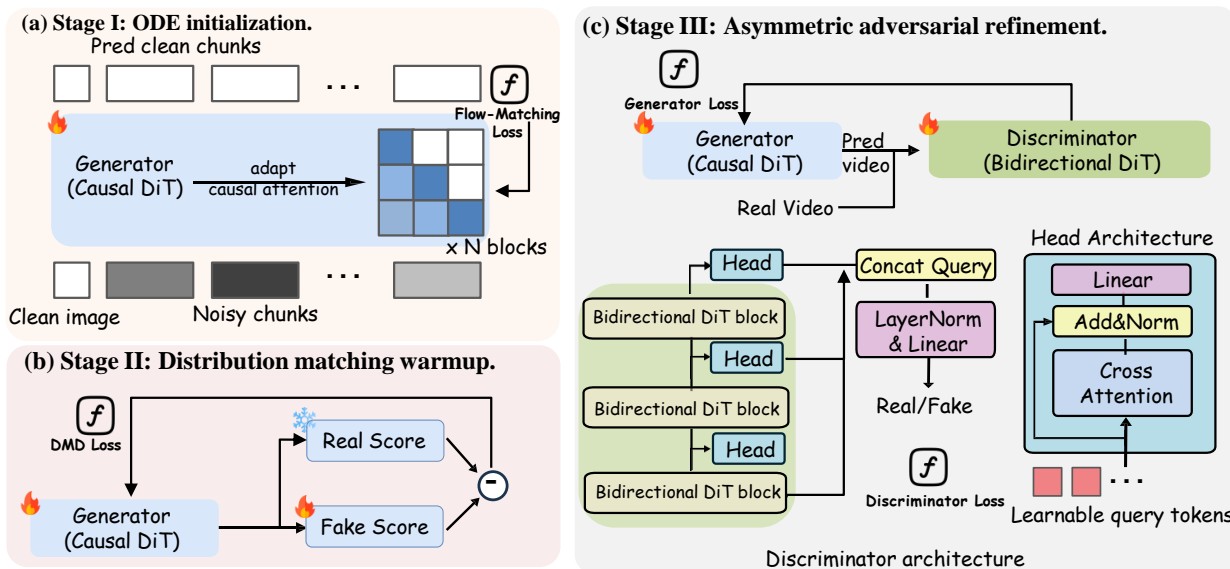

*Figure 3.* **Training Pipeline.** We train a one-step autoregressive generator $G_\theta$ through three stages. **(a) Stage I: ODE initialization** replaces bidirectional attention in pre-trained video models with block-wise causal attention, trained by diffusion-forcing with flow-matching loss. **(b) Stage II: One-step DMD Warmup** distills a strong diffusion teacher under self-rollout training by matching real and fake scores, bringing the student distribution close to the teacher. **(c) Stage III: Asymmetric Adversarial Refinement** autoregressively rolls out $G_\theta$ and trains it against a bidirectional discriminator. The discriminator uses bidirectional DiT blocks where a single group of learnable query tokens are used to aggregate full video context for video-level discrimination.

where $\delta \sim \mathcal{N}(0, \sigma^2 I)$ is a small perturbation applied at the same discriminator timestep $\tau$. The discriminator is optimized with $\mathcal{L}_D + \lambda \mathcal{L}_{\text{reg}}$. The bidirectional discriminator aggregates full video context through learnable query tokens, providing stronger temporal consistency signals including sensitivity to long-horizon drift.

**Rationale for staged training design.** Directly training an asymmetric setup (causal $G_\theta$ with a bidirectional $D_\psi$) is empirically unstable in the 1-step regime. The ODE and DMD stages move the student close to the teacher distribution, after which adversarial refinement can focus on improving visual quality and temporal coherence. Furthermore, since the teacher distribution and the real data distribution are inherently misaligned, adopting a DMD2-style joint DMD+GAN loss (Yin et al., 2024a) causes the two objectives to conflict: the DMD loss pulls the generator toward the teacher while the GAN loss pulls it toward real data, resulting in unstable training dynamics (Tong et al., 2025; Cheng et al., 2025). Separating them into sequential stages avoids this instability. We find this three-stage design crucial for stable training and high-quality results.

**Long-video generation mechanisms.** To enable stable infinite streaming, we adopt a Sink Token + Sliding Window attention mechanism (Xiao et al., 2023). We dedicate the first few tokens as "sink tokens" that always participate in attention to preserve global identity information, combined with a local sliding window for recent motion con-

text. Furthermore, we implement Relative RoPE (similar to StreamingLLM (Xiao et al., 2023)) to handle positional encoding extrapolation, ensuring that the relative distances between query and key embeddings remain within the training distribution regardless of the absolute frame index.

**Implementation details.** We employ the 14B Wan 2.1 T2V model as our backbone. For Image-to-Video (I2V), we encode the conditioning frame into the first KV cache position as a standalone chunk, while subsequent generation uses a chunk size of 4. We set the attention sink size to 1 and local window size to 9. Stages 1 and 2 follow Self Forcing (Huang et al., 2025). Specifically, we train the Stage 1 ODE model for 2,000 steps. In Stage 2, we set the update frequency ratio between the generator and the fake score model to 1:5. We train the DMD generator for only 100 steps and employ early stopping, as prolonged training empirically leads to motion collapse. In Stage 3, we initialize the discriminator with the Wan 2.1 T2V backbone and an APT-style head (Lin et al., 2025a), inserting cross-attention blocks at layers 19, 29, and 39. We utilize the approximated R1 and R2 regularizations as described in the Method section to stabilize the 14B model, setting the regularization weight $\lambda = 20$ with a perturbation scale of $\sigma_{\text{reg}} = 0.05$. Additionally, we apply timestep-dependent Gaussian noise to the discriminator inputs, sampling $\tau \sim \mathcal{U}[0, 1000]$ to match the generator's noise schedule. For the generator, we use a learning rate of $4 \times 10^{-7}$ with EMA

*Table 1.* **Quantitative comparison on VBench-I2V (Huang et al., 2024).** We compare our method against autoregressive baselines using 4-NFE sampling (CausVid (Yin et al., 2025) and Self Forcing (Huang et al., 2025)), and include the bidirectional model Wan 2.1 I2V (Wan et al., 2025) with 100-NFE sampling (50 steps with CFG guidance) as reference. Our model with full three-stage training achieves state-of-the-art performance among autoregressive methods using only a single sampling step. The **best** result in each column is shown in bold, and the second-best result is underlined.

| Method | Quality | | | | | | Condition | |
|---|---|---|---|---|---|---|---|---|
| | Subject Consistency↑ | Background Consistency↑ | Motion Smoothness↑ | Dynamic Degree↑ | Aesthetic Quality↑ | Imaging Quality↑ | I2V Subject↑ | I2V Background↑ |
| *Bidirectional* | | | | | | | | |
| Wan 2.1 I2V (100 NFE) | 93.88 | 94.86 | 98.14 | **51.09** | **64.97** | 70.12 | 96.80 | **98.59** |
| *Autoregressive* | | | | | | | | |
| CausVid (4 NFE) | 83.45 | 89.37 | **98.61** | 33.80 | 61.55 | 70.60 | 92.91 | 83.34 |
| Self Forcing (4 NFE) | 91.77 | 93.41 | 98.55 | 34.93 | 60.96 | **71.50** | 95.79 | 91.18 |
| Ours (1 NFE, Stage-II) | 92.14 | 92.13 | 98.04 | 50.30 | 58.64 | 69.37 | 96.56 | 95.12 |
| Ours (1 NFE, Stage-III) | **94.34** | **95.08** | 98.22 | 41.46 | 60.07 | 71.49 | **98.65** | 97.83 |

decay 0.98; for the discriminator, we do not apply EMA and set the backbone learning rate to $1 \times 10^{-6}$ and the head learning rate to $2 \times 10^{-6}$. We use a batch size of 256 via gradient accumulation for training stability and train the generator for 200 steps.

## 5. Experiments

We evaluate the effectiveness of our proposed Asymmetric Adversarial Distillation framework on large-scale video generation benchmarks. We focus on two key aspects: (1) the quality and stability of few-step streaming generation compared to autoregressive and diffusion baselines, and (2) the impact of discriminator architecture design on training stability and motion quality.

### 5.1. Comparison with State-of-the-Art Methods

We evaluate I2V short-video generation under the official VBench standard protocol, producing 5-second clips at a unified 480p resolution. We compare against representative diffusion and autoregressive baselines in Table 1, including Wan 2.1 (Wan et al., 2025), CausVid (Yin et al., 2025), and Self Forcing (Huang et al., 2025). For CausVid and Self Forcing, we follow their published evaluation settings and report zero-shot results. Table 1 reports per-aspect VBench metrics on both generation quality and conditioning faithfulness. Overall, our method achieves strong I2V conditioning performance and imaging quality. Figures 4 and 5 provide qualitative comparisons and user preferences, respectively.

As shown in Table 1, our one-step model achieves competitive generation quality compared to multi-step autoregressive baselines while requiring only a single forward pass. In particular, the Stage-III model achieves the best autoregressive performance in subject consistency (94.34), background consistency (95.08), and I2V subject faithfulness (98.65), while also reaching 97.83 on I2V background faithfulness

and 71.49 on imaging quality. Compared with CausVid and Self Forcing, our method substantially improves scene coherence and conditioning preservation, indicating that the proposed asymmetric adversarial distillation effectively stabilizes long-horizon generation. We also observe a clear trade-off between Stage-II and Stage-III training: Stage-II yields stronger motion magnitude (Dynamic Degree 50.30), whereas Stage-III provides better consistency and faithfulness overall. Figures 4 and 5 further support these findings: our method reduces identity drift and receives higher user preference scores in perceptual comparisons.

We further assess perceptual quality via a side-by-side user study on motion realism and image quality. Figure 5 shows that our method is preferred over both Self Forcing and CausVid, indicating stronger perceived quality.

### 5.2. Ablation Studies

We investigate optimal training strategies for one-step causal generation. We first examine the necessity of the stage-wise DMD training pipeline in Figure 6, and then ablate discriminator topology at the 14B scale to understand what forms of adversarial supervision lead to stable long-horizon motion. Finally, we analyze why a full-step causal teacher can be unreliable as supervision due to drift.

For the Causal Backbone settings in our ablation, we initialize the discriminator from the Stage 2 DMD-trained generator, ensuring both models start from the same distribution. We also enforce the exact same block-wise causal attention mask. Regarding the logit heads: for video-wise logits, the learnable query token performs cross-attention over the entire spatiotemporal sequence to aggregate global features; for frame-wise logits, the query token performs cross-attention restricted to individual frame tokens independently, lacking global temporal aggregation capabilities.

Prompt: a group of jellyfish swimming in an aquarium

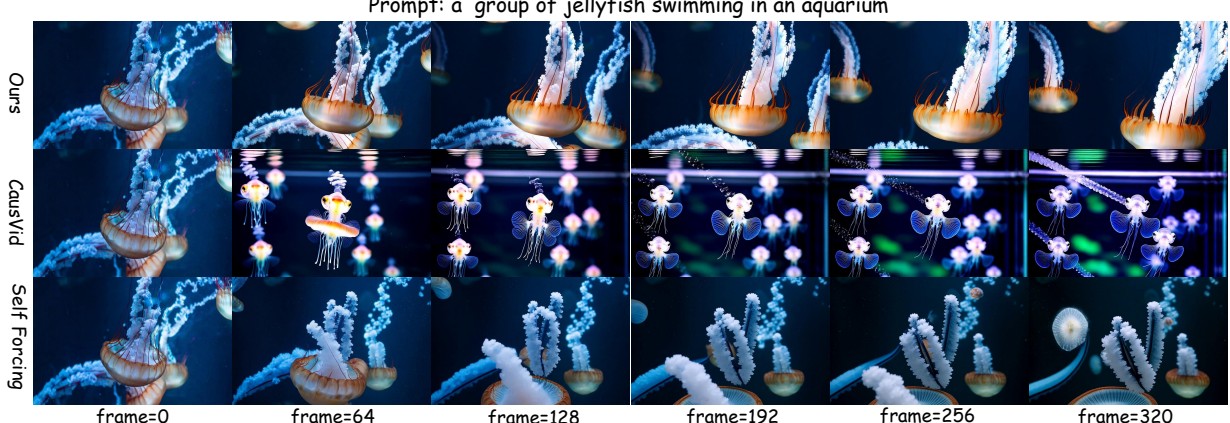

frame=0    frame=64    frame=128    frame=192    frame=256    frame=320

*Figure 4.* **Qualitative comparison.** We compare our method against autoregressive baselines using 4-NFE sampling (CausVid (Yin et al., 2025) and Self Forcing (Huang et al., 2025)). Given a conditioning image of a swimming jellyfish, our method synthesizes vivid motion while maintaining visual fidelity and identity consistency over long horizons (up to 320 frames), whereas baselines exhibit identity drift.

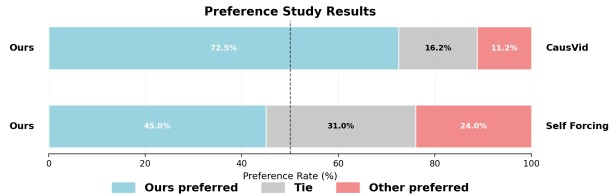

*Figure 5.* **User Preference Study.** Win rates of our method against baselines (Self Forcing, CausVid). Our method is preferred in the majority among these methods.

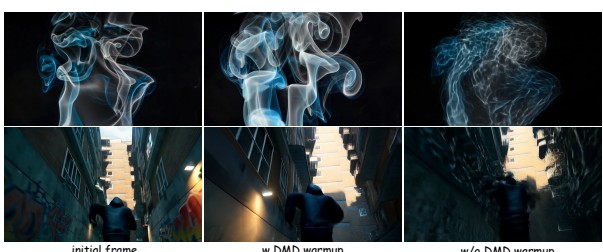

*Figure 6.* **Stage-wise ablation of DMD warmup.** DMD warmup helps stabilize subsequent adversarial refinement and prevents severe visual degradation.

**Ablation on DMD warmup.** We ablate the DMD warmup stage to verify whether adversarial refinement alone can reliably train a one-step autoregressive generator. As shown in Table 2 and Figure 6, removing DMD warmup leaves the initial generator distribution too far from the data distribution, making the subsequent GAN objective unstable and causing severe visual degradation. With DMD warmup, the generator starts from a much better one-step solution, preserving scene structure and object appearance before adversarial training improves temporal realism.

*Table 2.* **Ablation on DMD warmup.** DMD warmup improves one-step generation quality before adversarial refinement.

| Method | Aesthetic Quality ↑ | Imaging Quality ↑ |
|---|---|---|
| w/o DMD warmup | 53.63 | 62.81 |
| w/ DMD warmup | 58.64 | 69.37 |

**Analysis of discriminator architectures.** We systematically analyze the impact of discriminator topology (Table 3), with qualitative examples shown in Figure 7, through the lens of our theoretical proofs in Appendix A. The evaluation is conducted on 100 videos randomly sampled from the VBench-I2V benchmark and our dataset. We measure Dynamic Degree on 5-second videos, while Drift Score is evaluated on 20-second rollouts to better capture long-horizon error accumulation. The primary driver of performance is the backbone's causality. As proven in Proposition A.1, causal discriminators effectively suffer from linear error accumulation. A causal backbone prevents the future-anchored gradients necessary to critique early decisions based on global outcomes (Proposition A.2). For causal backbones, granularity is critical: frame-wise heads produce completely static videos (Dynamic Degree 1.08), while video-wise heads restore motion (42.07) but still exhibit severe drift. We attribute the motion collapse in frame-wise discrimination to a trivial solution: since the discriminator only evaluates the marginal distribution $p(x_t)$ of each frame independently, and any previous frame $x_{t-1}$ is itself a perfectly realistic image, the generator can achieve a high discriminator score by simply copying $G(x_{<t}) = x_{t-1}$, producing static video. Video-wise heads avoid this failure mode by enforcing temporal coherence across the sequence.

For bidirectional backbones, both granularity settings perform comparably, with video-wise logits achieving slightly

Prompt: Drone wide-angle flyover of Cancun beach at sunset, orange–pink–purple sky reflecting on turquoise water above golden sand.

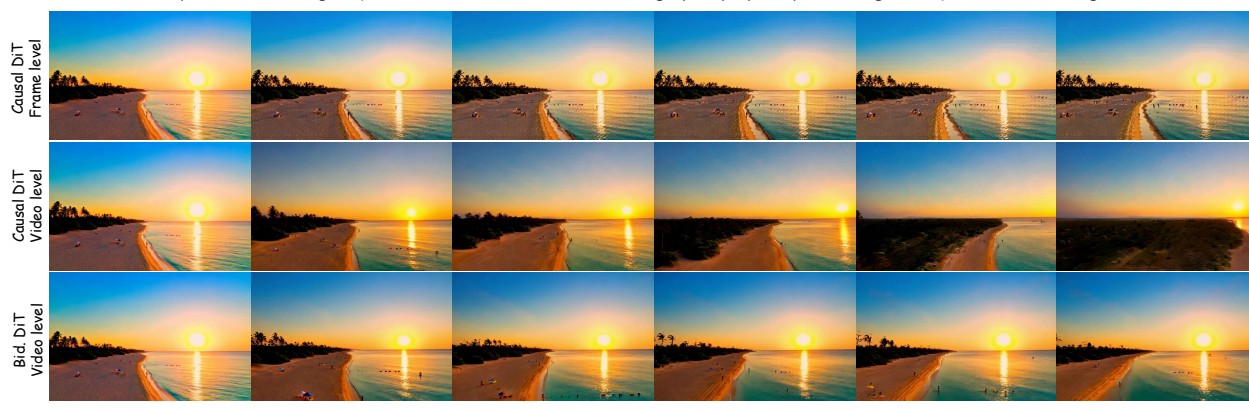

Prompt: A brown horse is trotting along a dirt path leading towards a small village surrounded by rolling hills and lush green fields.

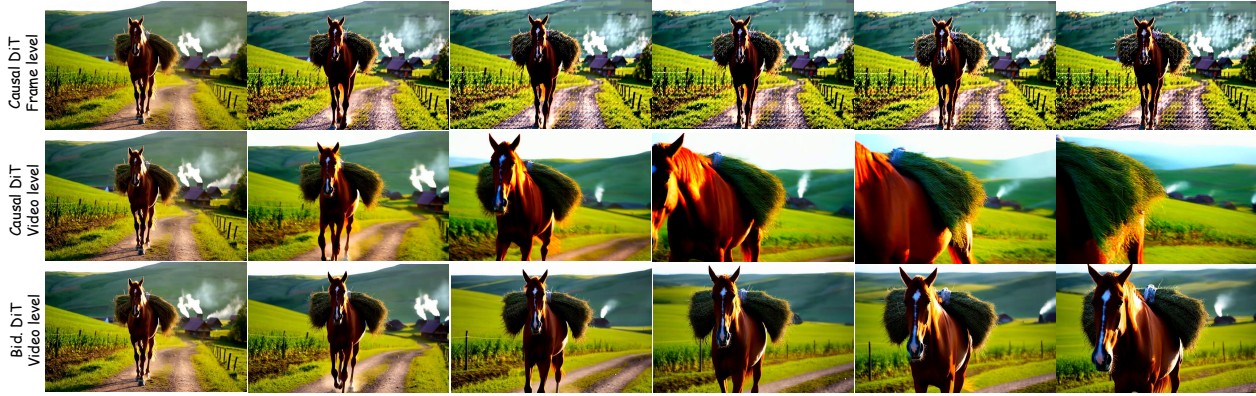

*Figure 7.* **Qualitative ablation study.** We compare generated motion under four settings: (a) **Causal backbone w/ frame-wise logits** results in completely static videos; (b) **Causal backbone w/ video-wise logit** and (c) **Bidirectional backbone w/ frame-wise logits** are both prone to drift, exhibiting erratic camera movement, excessive speed, or color shifts. **(d) Bidirectional backbone w/ video-wise logit (Ours)** achieves the best performance with stable generation.

*Table 3.* **Ablation on Discriminators.** We compare Causal vs. Bidirectional visibility and Frame-wise vs. Video-wise granularity. Causal + Frame-wise produces completely static videos (Dynamic Degree 1.08); Causal + Video-wise has high dynamics but severe drift. Bidirectional backbones provide stable supervision, with Video-wise logits achieving the best drift mitigation.

| Backbone | Logit Granularity | Drift Score↓ | VBench Dynamics↑ |
|---|---|---|---|
| Causal DiT | Frame-wise | N/A | 1.08 |
| Causal DiT | Video-wise | 7.10 | **42.07** |
| Bidirectional DiT | Frame-wise | 4.38 | 39.04 |
| Bidirectional DiT | Video-wise | **4.02** | 39.29 |

better drift mitigation (4.02 vs. 4.38). We hypothesize that bidirectional attention already enables deep feature interaction across the entire spatiotemporal volume within the bidirectional DiT backbone, which makes the head's aggregation strategy less critical.

**Drift in a full-step causal teacher.** To isolate the limitations of causal supervision itself, we construct a full-step

causal teacher by adapting a Wan 2.1 T2V model (Wan et al., 2025) into a causal generator using the 1.3B variant. Specifically, we replace bidirectional attention with a block-wise causal mask, allowing tokens within a frame chunk to attend bi-directionally while preventing attention to future chunks. We train this causal teacher using Diffusion Forcing (Chen et al., 2024), which conditions the current chunk's denoising process on noisy versions of previous chunks to bridge the train–test gap. At inference time, the model generates videos autoregressively in a chunk-wise manner.

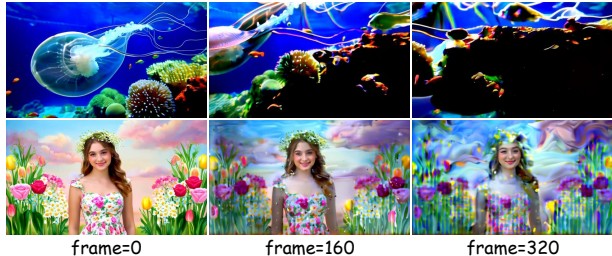

frame=0        frame=160        frame=320

*Figure 8.* **Drift in Causal Video Diffusion Model.** Long-horizon rollout from the full-step causal teacher.

However, even when this full-step causal teacher converges, we observe severe autoregressive error accumulation: long-horizon rollouts exhibit geometric distortion and identity loss (Figure 8), suggesting a drifting distribution $p_{\text{drift}}(x_{1:T})$. Using such a drifting causal teacher directly as a discriminator $D(x_{1:T})$ can therefore provide flawed supervision, since the drifting trajectory remains high-likelihood under the teacher itself. This motivates our asymmetric adversarial distillation with a bidirectional discriminator that can provide future-anchored critiques.

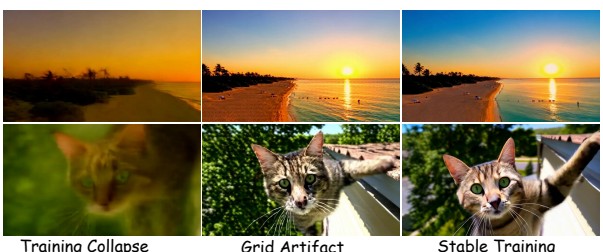

Figure 9. **Effect of regularization coefficient** $\lambda$. Without regularization ($\lambda = 0$), training collapses. Excessive regularization ($\lambda = 50$) introduces grid-like patterns. The optimal setting ($\lambda = 20$) balances stability and visual quality.

**Analysis of regularization coefficient.** Beyond architectural choices, we find that the regularization coefficient $\lambda$ plays a critical role in training stability. As illustrated in Figure 9, setting $\lambda = 0$ (i.e., removing the regularization term entirely) leads to rapid training collapse, where the generator produces degenerate outputs. Conversely, an overly large coefficient ($\lambda = 50$) introduces visible grid-like artifacts in the generated frames, likely due to over-regularization suppressing fine-grained texture details. We empirically find that $\lambda = 20$ strikes a good balance, maintaining stable adversarial training while preserving visual fidelity.

## 6. Conclusion

We proposed AAD-1, an asymmetric adversarial distillation framework for one-step autoregressive video generation. By employing a bidirectional discriminator with video-level holistic discrimination and a phased training strategy with distribution matching warm-up, AAD-1 effectively addresses motion collapse and training instability. Extensive experiments on VBench demonstrate that AAD-1 achieves state-of-the-art performance with superior visual quality and motion fidelity. We hope our work provides valuable insights for efficient autoregressive video generation.

## Limitations

Despite its strong chunk-wise one-step autoregressive generation, our method has limitations in fast motion, complex structures, and long-horizon extrapolation.

**Fast motion.** The one-step setting can struggle in fast-moving scenes, where large inter-frame motion must be predicted by a single denoising pass rather than refined across multiple sampling steps. In such cases, we observe blurry frames, distorted structures, or degraded temporal coherence, reflecting the difficulty of compressing iterative diffusion sampling into very few steps (Yin et al., 2024b; Lin et al., 2025a). Improving one-step objectives for large motion remains important for robust streaming generation.

**Complex structures.** Compared with APT2-style one-step-per-image generation (Lin et al., 2025b), where each step can focus on local synthesis for a single image, our chunk-wise one-step setting requires the generator to synthesize multiple latent frames within a chunk in a single forward pass. This makes preserving fine-grained details and subtle local dynamics more challenging, especially for complex and highly structured content such as human faces and hands. These challenges suggest a need for training objectives and generation strategies that better capture complex local structure under chunk-wise one-step generation.

**Long-horizon extrapolation.** Our adversarial refinement is trained on 5-second clips due to data and compute constraints, as high-quality long-video training data remains scarce and expensive to curate. Although the model can extrapolate beyond this horizon, long rollouts may exhibit drift and quality degradation as errors accumulate over autoregressive chunks, consistent with long-horizon autoregressive video generation challenges (Lin et al., 2025b). We hypothesize that longer-video adversarial training could alleviate this issue by exposing the generator to long-range temporal failures and accumulated rollout errors.

## Acknowledgements

This work was supported in part by the Natural Science Foundation of China under Grant No. 62503323, the Ant Group Research Intern Program, and the Ant Group Postdoctoral Programme.

## Impact Statement

This paper presents work whose goal is to advance the field of Machine Learning, specifically in efficient video generation. While our method enables faster autoregressive video synthesis, we acknowledge potential dual-use concerns common to generative models, including the creation of misleading or harmful content. We encourage the development of detection mechanisms and responsible deployment practices alongside this technology. There are many potential societal consequences of our work, none which we feel must be specifically highlighted here beyond these standard considerations for generative AI systems.

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

# A. Theoretical Analysis of Ablation Settings

**Notation.** Let $x_{1:T}$ denote a video clip (conditioned on context $c$). We denote the data distribution by $p(x_{1:T})$ and the causal generator's rollout distribution by $q(x_{1:T})$. Let $x_{<t} \triangleq x_{1:t-1}$. For distributions $P, Q$ with densities $p, q$, the KL divergence is $\mathrm{KL}(P\|Q) \triangleq \mathbb{E}_{x \sim P}[\log(p(x)/q(x))]$.

## A.1. On-Policy Error Accumulation in Causal Rollouts

**Proposition A.1** (Linear Error Accumulation). *Let $p(x_{1:T}) = \prod_{t=1}^{T} p_t(x_t \mid x_{<t})$ and $q(x_{1:T}) = \prod_{t=1}^{T} q_t(x_t \mid x_{<t})$ be two autoregressive distributions. If the expected on-policy conditional KL divergence is bounded by $\varepsilon$ at each step, i.e.,*

$$\forall t, \quad \mathbb{E}_{x_{<t} \sim q}\Big[\mathrm{KL}\Big(q_t(\cdot \mid x_{<t}) \,\Big\|\, p_t(\cdot \mid x_{<t})\Big)\Big] \leq \varepsilon, \tag{11}$$

*then the joint KL divergence satisfies $\mathrm{KL}(q(x_{1:T})\|p(x_{1:T})) \leq T\varepsilon$.*

*Proof.* We expand the KL divergence definition using the chain rule for autoregressive models.

$$\mathrm{KL}(q\|p) = \int q(x_{1:T}) \, \log \frac{\prod_{t=1}^{T} q_t(x_t \mid x_{<t})}{\prod_{t=1}^{T} p_t(x_t \mid x_{<t})} \, dx_{1:T}$$

$$= \sum_{t=1}^{T} \int q(x_{1:T}) \, \log \frac{q_t(x_t \mid x_{<t})}{p_t(x_t \mid x_{<t})} \, dx_{1:T}.$$

Consider the $t$-th term in the summation. We decompose $q(x_{1:T}) = q(x_{<t}) q_t(x_t \mid x_{<t}) q(x_{>t} \mid x_{\leq t})$ and integrate out the future variables $x_{>t}$:

$$\int q(x_{1:T}) \log \frac{q_t(x_t \mid x_{<t})}{p_t(x_t \mid x_{<t})} \, dx_{1:T}$$

$$= \int q(x_{<t}) \left[ \int q_t(x_t \mid x_{<t}) \log \frac{q_t(x_t \mid x_{<t})}{p_t(x_t \mid x_{<t})} \, dx_t \right] dx_{<t}$$

$$= \mathbb{E}_{x_{<t} \sim q}\Big[\mathrm{KL}\big(q_t(\cdot \mid x_{<t})\|p_t(\cdot \mid x_{<t})\big)\Big].$$

Substituting this back into the sum and applying the bound from Eq. (11), we obtain:

$$\mathrm{KL}(q\|p) = \sum_{t=1}^{T} \mathbb{E}_{x_{<t} \sim q}[\mathrm{KL}_t] \leq \sum_{t=1}^{T} \varepsilon = T\varepsilon.$$

$\square$

*Remark.* This result highlights that controlling the *one-step* error $\varepsilon$ on the *generator's own induced distribution* (on-policy matching) is sufficient to bound the sequence-level drift linearly in $T$. Our Stage III self-rollout training explicitly targets this on-policy minimization.

## A.2. Analysis of backbone visibility

**Proposition A.2** (Future-Anchored Gradients in Bidirectional Backbones). *Let $s_t(x_{1:T}) = \mathrm{Head}(H_t)$ be the discriminator logit for frame $t$, where $H_t$ is the backbone representation.*

1. **Causal backbone:** *If the backbone is causal, $H_t$ depends only on $x_{\leq t}$. Thus, $\frac{\partial s_t}{\partial x_{>t}} = 0$.*

2. **Bidirectional backbone:** *If the backbone is bidirectional, $H_t$ depends on $x_{1:T}$. Thus, in general, $\frac{\partial s_t}{\partial x_{>t}} \neq 0$.*

*Proof.* **Case (i): causal backbone.** A causal backbone enforces a mask $M_{ij} = 0$ for $j > i$. The representation $H_t$ at index $t$ is computed as a function of inputs $x_1, \ldots, x_t$ only. Formally, $H_t = f_t(x_{\leq t})$. For any suffix variation $x'_{>t} \neq x_{>t}$, we

have $H_t(x_{\leq t}, x_{>t}) = H_t(x_{\leq t}, x'_{>t})$, implying $s_t$ is invariant to future frames. Consequently, gradients cannot propagate from future content violations back to time $t$.

**Case (ii): bidirectional backbone.** A bidirectional backbone allows attention to all tokens. The representation is a function of the full sequence: $H_t = g_t(x_{1:T})$. A perturbation in the future $x_{>t}$ alters $H_t$ via the attention mechanism, changing $s_t$. By the chain rule, $\frac{\partial s_t}{\partial x_{>t}} = \frac{\partial s_t}{\partial H_t} \frac{\partial H_t}{\partial x_{>t}}$, which is non-zero. This mechanism allows the discriminator to act as an "anchor," penalizing step $t$ if it is inconsistent with the (ground-truth) future $x_{>t}$ provided during offline training. $\square$

*Note on causal backbone with a video-wise head.* In the setting with a causal backbone and a video-wise head, the final score $S = \text{Pool}(\{s_t\}_{t=1}^T)$ depends on all frames. However, the feature extraction $H_t$ remains causal. The future dependency is "late fusion" (gradients flow from $S$ to $H_t$ based on pooling weights, but $H_t$ itself does not contain future features). In contrast, a bidirectional backbone provides "early fusion," enriching $H_t$ with future context directly.

### A.3. Analysis of logit granularity

**Proposition A.3** (Video-wise Heads Subsume Frame-wise Heads). *Let the backbone outputs be $H = [H_1, \ldots, H_T]$. A frame-wise head queries only $H_t$ to score frame $t$, while a video-wise head queries $H_{1:T}$. The class of functions implementable by video-wise heads strictly includes those implementable by frame-wise heads.*

*Proof.* Consider a standard attention mechanism $\text{Attn}(Q, K, V)$. The frame-wise head for frame $t$ computes $y_t^{\text{frame}} = \text{Attn}(Q_t, H_t W_K, H_t W_V)$. The video-wise head computes $y^{\text{video}} = \text{Attn}(Q_{\text{global}}, HW_K, HW_V)$ with a mixing mask $M$. We can emulate the frame-wise behavior in the video-wise architecture by constructing a block-diagonal mask $M$ in the video-wise head such that query tokens corresponding to time $t$ can only attend to keys at time $t$ (setting $M_{i,j} = -\infty$ if $tokens(i) \in t, tokens(j) \notin t$). Under this masking, the softmax normalizes only over single-frame tokens, recovering the exact computation of the frame-wise head (assuming shared weights). Since the video-wise head can instantiate this block-diagonal masking pattern while also allowing cross-frame attention patterns, it is strictly more expressive. $\square$

## B. Additional Quantitative Results

**Drift score.** Following Reward Forcing (Lu et al., 2025b), we quantify long-horizon visual drift by computing the standard deviation of imaging-quality scores along the temporal horizon. Specifically, we evaluate imaging quality over temporal segments of each long rollout and average the resulting standard deviation across videos. A lower Drift Score indicates more stable visual quality over time.

We provide additional quantitative results on VBench-I2V (Huang et al., 2024) to complement the main paper. Beyond the standard 1-NFE, 480p, 5-second setting, we evaluate a 2-NFE variant, 20-second rollouts, and zero-shot 720p generation.

The 2-NFE variant is included as an inference-budget reference. It uses the same three-stage training pipeline as AAD-1, including ODE initialization, DMD warmup, and asymmetric adversarial refinement. For adversarial stabilization, we follow Self Forcing (Huang et al., 2025) and add timestep-dependent Gaussian noise to the discriminator inputs. For generated rollouts corresponding to a given generator output timestep, the discriminator noise level is sampled from the associated timestep interval, keeping the noised discriminator inputs consistent with the generator's output distribution. As shown in Table 4, the slightly larger sampling budget improves motion smoothness and dynamic degree while maintaining strong I2V subject and background faithfulness.

The 20-second and 720p settings are evaluated in a zero-shot manner from the standard AAD-1 model, without additional training on longer videos or higher-resolution data. These results help illustrate how different inference settings affect temporal consistency, motion dynamics, visual quality, and image-to-video condition preservation.

## C. Training Cost and Memory

We provide additional details on the training cost and memory footprint of our method. Full training takes approximately 3.5 days on 64 NVIDIA H20 GPUs, including about 0.5 day for Stage I, 1 day for Stage II, and 2 days for Stage III. To reduce memory usage, we employ Ulysses-style context parallelism (Jacobs et al., 2023) with context parallel size 8 together with PyTorch activation checkpointing. Under the same Stage III setup, namely 64 H20 GPUs, 8 GPUs per node, and Ulysses-style context parallelism with $\text{cp} = 8$, the bidirectional discriminator adversarial training reaches a peak total GPU

*Table 4.* **Additional quantitative results on VBench-I2V (Huang et al., 2024).** Wan 2.1 I2V (Wan et al., 2025), sampled with 100 NFE, is included as a bidirectional reference. All AAD-1 variants are evaluated under different inference settings.

| Method | Setting | Quality | | | | | | Condition | |
|---|---|---|---|---|---|---|---|---|---|
| | | Subject Consistency↑ | Background Consistency↑ | Motion Smoothness↑ | Dynamic Degree↑ | Aesthetic Quality↑ | Imaging Quality↑ | I2V Subject↑ | I2V Background↑ |
| *Bidirectional reference* | | | | | | | | | |
| Wan 2.1 I2V | 100 NFE | 93.88 | 94.86 | 98.14 | 51.09 | 64.97 | 70.12 | 96.80 | 98.59 |
| *AAD-1 variants* | | | | | | | | | |
| AAD-1 | 480p, 5s, 1 NFE | 94.34 | 95.08 | 98.22 | 41.46 | 60.07 | 71.49 | 98.65 | 97.83 |
| AAD-1 | 480p, 5s, 2 NFE | 94.03 | 95.52 | 98.99 | 50.04 | 59.46 | 71.00 | 98.06 | 98.50 |
| AAD-1 | 480p, 20s, 1 NFE | 84.31 | 89.30 | 98.93 | 60.98 | 55.48 | 68.61 | 97.43 | 97.25 |
| AAD-1 | 720p, 5s, 1 NFE | 94.52 | 95.63 | 98.76 | 24.39 | 61.03 | 72.29 | 98.30 | 98.70 |

memory usage of approximately 1040 GB and requires about 49 hours of training, while the causal discriminator adversarial training baseline uses approximately 830 GB and requires about 65 hours. The bidirectional discriminator incurs a higher memory cost because it processes the full sequence jointly; however, it can exploit FlashAttention-3 (Shah et al., 2024) for efficient full-sequence attention, whereas the causal discriminator relies on FlexAttention to implement causal masking, which results in slower training in practice.

## D. Inference Efficiency

We report latency and throughput on a single H100 GPU following the Self-Forcing protocol. Since runtime depends strongly on model size, we compare 1 NFE and 4 NFE inference at matched parameter scales. As shown in Table 5, reducing the sampling budget from 4 NFE to 1 NFE consistently lowers latency and improves throughput within each scale.

*Table 5.* **Inference efficiency.** Latency and throughput are measured on a single H100 GPU.

| NFE | 1.3B | | 14B | |
|---|---|---|---|---|
| | Latency (s)↓ | Throughput (FPS)↑ | Latency (s)↓ | Throughput (FPS)↑ |
| 1 | 0.289 | 43.37 | 1.134 | 14.33 |
| 4 | 0.714 | 17.70 | 2.822 | 5.71 |

