# OpenReview forum: "AAD-1: Asymmetric Adversarial Distillation for One-Step Autoregressive Video Generation"
_ICML.cc/2026/Conference — ICML 2026 regular_

### Official Review · Reviewer_coQJ · 2026-02-21

**Soundness:** 3
**Presentation:** 3
**Significance:** 3
**Originality:** 3
**Overall Recommendation:** 5
**Confidence:** 4

**Summary:**

This paper presents AAD-1, a post-training framework for one-step autoregressive image-to-video generation. The key idea is asymmetric adversarial distillation and a phased training pipeline. They report strong results on VBench and show qualitative improvements over prior baselines.

**Compliance With Llm Reviewing Policy:**

Affirmed.

**Key Questions For Authors:**

1) Training sensitivity and reproduction: GAN is well-known to be unstable in training, please add more analyses on the training sensitivity and reproduction. For example, besides λ, how sensitive is training to (a) the discriminator noise schedule, (b) R1/R2 settings, and (c) the D/G update ratios?

2) Generalization to more settings: Can you provide a more systematic eval covering different video durations and resolutions, as well as different model sizes and backbones?

**Limitations:**

yes

**Strengths And Weaknesses:**

● Soundness: (pros) The asymmetric discriminator design is well-motivated. The phased training recipe also makes sense for stability. The ablations improved the faithfulness of the experiments.  (cons) many stabilizers (noise injection to D, regularization, λ tuning, multi-stage training) are necessary but it’s hard to tell how robust this is across datasets/backbones/computes. Also, the evaluation test set is composed of VBench 480p 5-seconds clips which is far from real application scenarios; I’d like comparisons using longer and higher-resolution videos.

● Presentation: (pros) The paper is pretty readable: it clearly states the problems and ties them to the proposed solutions.  (cons) Some implementation choices are buried in text, and reproducibility would benefit from a tighter “default config” table.

● Significance: If this generalizes, one-step per frame autoregressive video generation is a big deal for streaming / real-time applications.

● Originality: The components individually aren’t brand new, they are practically motivated combination for the problem setting.

---

> ### Author Rebuttal · Authors · 2026-03-31
>
> We appreciate the reviewer’s acknowledgment of **“well-motivated design, faithful experiments, clear statements, big deal for streaming/real-time applications when it generalizes and practically motivated problem setting”.** We address the remaining concerns below.
>
> **Q1. Training sensitivity and reproduction.**
>
> **A1:** Following the reviewer’s suggestion, we conducted additional ablations on key training hyperparameters, including the discriminator noise schedule, R1/R2 regularization, and the G/D update ratio. N/A indicates training collapse, where the model fails to produce meaningful videos.
>
> | setting | Image quality | dynamic degree |
> | --- | --- | --- |
> | Ours | 70.42 | 45.35 |
> | w/o R1 | N/A | N/A |
> | w/o R2 | 70.10 | 40.22 |
> | w/o discriminator noise schedule | 69.33 | 10.82 |
> | G/D ratio = 2 | N/A | N/A |
>
> These results show that training behavior is systematic rather than arbitrary. R1 is essential for stability, as removing it consistently leads to collapse. The discriminator noise schedule mainly affects temporal dynamics, with a substantial drop in dynamic degree when removed. The G/D update ratio is also sensitive, with higher ratios (e.g., 2) causing instability. In contrast, R2 has a milder effect.
>
> Overall, while adversarial training is known to be sensitive, our results indicate that the final configuration is a stable and reproducible operating regime, with each component playing a distinct role.
>
> **Q2. Generalization to more settings.**
>
> **A2.** To further demonstrate long-horizon generation beyond the 5-second clips commonly used in prior work, we include 10s and 20s examples for both dynamic objects and scenes in the supplementary material. We will release code and models to facilitate reproducibility.
>
> Following the reviewer’s suggestion, we also evaluate our model under varying video durations and output resolutions:
>
> |  | subject consistency | background consistency | motion smoothness | dynamic degree | aesthetic quality | imaging quality | I2V subject | I2V background |
> | --- | --- | --- | --- | --- | --- | --- | --- | --- |
> | Ours 80 frame | 93.67 | 94.44 | 98.70 | 45.35 | 59.70 | 70.42 | 96.12 | 92.88 |
> | Ours 320 frame | 85.41 | 85.10 | 98.66 | 69.71 | 56.71 | 67.88 | 94.94 | 91.63 |
> | Ours-720p | 93.13 | 94.29 | 98.70 | 41.03 | 59.89 | 70.19 | 96.30 | 92.83 |
>
> These results indicate consistent behavior across both duration and resolution. Compared to the default setting, 320-frame videos maintain similar motion smoothness and achieve higher dynamic degree, while showing expected degradation in subject/background consistency and image quality under longer temporal horizons. At 720p, performance remains close to the default setting across metrics, suggesting good scalability with resolution.

---

> > ### Author Rebuttal · Reviewer_coQJ · 2026-04-01
> >
> > my concerns are mostly addressed

---

> > > ### Author Response · Authors · 2026-04-02
> > >
> > > Thank you very much for the thoughtful follow-up and for carefully considering our rebuttal. We are very glad that our additional analyses helped address your concerns.
> > >
> > > We also sincerely appreciate your constructive suggestions on training sensitivity, reproducibility, and evaluation under longer durations and higher resolutions. They helped us strengthen both the empirical support and the presentation of the paper.
> > >
> > > Thank you again for your careful evaluation and supportive feedback throughout the review process.

---

### Official Review · Reviewer_u6aq · 2026-03-09

**Soundness:** 3
**Presentation:** 4
**Significance:** 4
**Originality:** 2
**Overall Recommendation:** 4
**Confidence:** 5

**Summary:**

The work explores different discriminator architectural designs for the adversarial post-training of autoregressive video generation models. Prior work APT2 uses symmetrical architecture design where both the generator and the discriminator are causal. This work compares causal discriminator and bidirectional discriminator and finds that a bidirectional discriminator yields better results.

**Compliance With Llm Reviewing Policy:**

Affirmed.

**Final Justification:**

I am supporting the acceptance of the paper. The research direction in autoregressive streaming video generation is a very relevant topic today.

**Key Questions For Authors:**

1. Why is APT2 not included in Table 1? It seems that APT2, though close source, are evaluated under the same setting. APT2 seems to achieve better scores, which could be due to the differences in base model. This can be noted. But I think is is an over claim of "achieving state-of-the-art performance among autoregressive methods using only a single sampling step."

2. In table 2, the paper concluded that using a causal discriminator with frame-level logit granularity result in videos with no motion and a VBench dynamics score of 1.08. But APT2 uses a causal discriminator with frame-level logit but reports a VBench dynamic score of 42.44. This contradicts with the results in this research. Could it be implementation errors?

**Limitations:**

No intrinsic limitations to the proposed method. The main contribution of the method is on discriminator architecture. The use of multi-staged distillation, though inconvenient, is common among autoregressive video distillation work and not is introduced by this paper.

**Strengths And Weaknesses:**

1. Motivation is sound and the technique proposed are sound.
2. The presentation is clear. The background and methodology are well conveyed.
3. The research direction in autoregressive streaming video generation is a very relevant topic today.
4. Originality is slightly lacking. The method is not very innovative. More of an extended research on top of prior work.

---

> ### Author Rebuttal · Authors · 2026-03-31
>
> Thank you for the positive assessment, including **“sound motivation and technique, clear presentation and relevant topic.”** We address the remaining concerns below.
>
> **Q1: Comparison with APT2 in Table 1.**
>
> **A1:** Thank you for the suggestion. APT2 is indeed highly relevant. We did not include it in Table 1 because it is not open-sourced, and its reported results are based on a different base model (an internal 8B model) as well as a different training dataset and pipeline, which makes a controlled comparison difficult.
>
> Table 1 is designed to compare methods under a consistent setting with the same base model and training data, so that the effect of different design choices can be isolated. Under this setup, incorporating APT2 results would not be directly comparable. In our attempts, reproducing an APT2-style pipeline on top of Wan 2.1 was also not stable, likely due to its sensitivity to base model and data alignment, which further makes a fair comparison challenging.
>
> Regarding the wording, we agree that it can be made more precise. In the final version, we will clarify that the claim is made within this controlled setting, and explicitly note that APT2 reports strong results under a different setup.
>
> **Q2: Comparison with APT2-style discriminator in Table 2.**
>
> **A2:** We do not believe this discrepancy is due to an implementation error, but rather due to differences in the overall training pipeline.
>
> Our conclusion in Table 2 is specific to the setting without additional mechanisms such as recycling inputs and heavy diffusion adaptation used in APT2. For a fair comparison in Table 2, we keep the training data and overall setup identical to our method, varying only the discriminator design to isolate the effect of causal (symmetric) versus asymmetric formulations. In this setting, a causal discriminator with frame-level logits tends to lead to motion collapse. As analyzed in Sec. 5.2, this is because the discriminator evaluates per-frame marginals $p(x_t)$, allowing the generator to minimize the loss by copying $x_{t-1}$, resulting in limited motion.
>
> Our observations are consistent with APT2. As described in the APT2 paper, it “recycling generated frames as inputs for the next autoregressive step,” which introduces additional temporal dependence to drive motion. Their project page also shows failure cases with limited motion and related analysis when such mechanisms are absent, aligning with our observations.
>
> Therefore, our findings do not contradict APT2, but instead highlight that the causal discriminator with frame-level logits alone is insufficient in this controlled setting.
>
> We will clarify this distinction in the final version to avoid confusion.

---

> > ### Author Rebuttal · Reviewer_u6aq · 2026-04-01
> >
> > Thanks for the response. I will remain my score.
> >
> > The main reason for the weak accept is that the conclusion on discriminator architecture is not generalizable. This limits the actual contributions of the work.

---

> > > ### Author Response · Authors · 2026-04-02
> > >
> > > Thank you very much for the thoughtful follow-up and for carefully considering our rebuttal. We are glad that our responses helped clarify our technical setting, and we sincerely appreciate your constructive feedback throughout the review process.
> > >
> > > We also appreciate your point regarding the generalizability of the conclusions on discriminator architecture. While this limitation is inherent to the scope of the current study, we hope to improve the paper by making the scope of our claims clearer and by better emphasizing that our findings are specific to the 1-step causal video distillation setting, rather than a universal statement across all adversarial video generation regimes.
> > >
> > > Thank you again for your careful evaluation and helpful suggestions.

---

### Official Review · Reviewer_8zm4 · 2026-03-12

**Soundness:** 2
**Presentation:** 3
**Significance:** 3
**Originality:** 2
**Overall Recommendation:** 4
**Confidence:** 3

**Summary:**

This paper addresses two major challenges in One-step autoregressive video generation. First, current methods often use a causal discriminator that is symmetric to the causal generator for frame-wise discrimination. Since this discriminator cannot see future frames, it fails to detect long-range temporal degradation. This leads to motion collapse where the video stays stuck on the first frame. Second, starting adversarial distillation from scratch causes early results to drift from the real data distribution. This error accumulates during self-unrolling training and leads to training instability / collapse.

To solve these, the authors propose a Bidirectional discriminator with video-level holistic discrimination. This allows the model to use the full spatial-temporal context to detect motion issues. The training also includes a Stage II Warm-up using Distribution Matching Distillation (DMD). This ensures the generator stays on-manifold before the adversarial stage begins.

**Compliance With Llm Reviewing Policy:**

Affirmed.

**Final Justification:**

Based on the original idea of ​​the paper, the experimental results, and the experimental results during the subsequent rebuttal, I support the acceptance of the paper.

**Key Questions For Authors:**

1. The paper identifies APT2 as a very relevant core competitor. Although you mentioned it relies on closed-source models, could you implement a baseline using the Wan 2.1 backbone with an APT2 style symmetric causal discriminator? This would clearly prove the benefit of your asymmetric design.

2. A Bidirectional discriminator with a global field of view is introduced to capture Long-range drift. However, using a 14B Wan 2.1 T2V backbone as a bidirectional discriminator to evaluate entire video sequences must involve massive VRAM and computational overhead. Could you explicitly report the GPU memory consumption and training time (GPU hours) for Stage III adversarial fine-tuning, as well as the additional overhead compared to using a standard causal discriminator?

3. The paper mentions that Stage II DMD warm-up was limited to only 150 steps followed by early stopping, stating that prolonged training empirically leads to severe motion collapse. This hyperparameter appears extremely sensitive and fragile. Why was 150 steps chosen as the specific threshold? How is the critical point of collapse defined and monitored? Furthermore, could you provide a Sensitivity analysis regarding how the number of Stage II training steps affects the final generation quality in Stage III?

**Limitations:**

Please refer to the questions above. I am willing to increase my score if the authors can adequately address these concerns.

**Strengths And Weaknesses:**

The results show that Subject Consistency reached 93.67 and background consistency reached 94.44. Both I2V Subject (96.12) and I2V Background (92.88) performed better than the baselines. While one-step models often produce static videos, AAD-1 achieved a Dynamic Degree of 45.35. This is much higher than the 33.80 and 34.93 seen in other models.

The work builds on Diffusion Forcing for initialization and uses Self-Forcing DMD for the one-step warm-up. The training stability techniques also draw from recent work like APT. Although some components are known, the asymmetric architecture of causal generation plus bidirectional video-level discrimination is a new contribution. The decoupled training pipeline effectively leads to SOTA performance in this field.

---

> ### Author Rebuttal · Authors · 2026-03-31
>
> Thank you for your time and for acknowledging our strong results in subject and background consistency and dynamic degree. We address the remaining concerns below.
>
> **Q1: Comparison with an APT2-style discriminator using the Wan 2.1 backbone.**
>
> **A1:** While exact replication of APT2 is not feasible due to differences in the base model (APT2 uses an internal 8B model) and unavailable training data, we implement a strong APT2-style baseline on Wan 2.1 with carefully tuned hyperparameters. For a fair comparison, we keep the training data and overall setup identical to our method, varying only the discriminator design to isolate the effect of causal (symmetric) versus asymmetric formulations.
>
> As analyzed in Lines 380–460 (Section 5.2, *Theoretical Analysis of Discriminator Architectures*), a causal discriminator with frame-wise logit supervision tends to induce motion collapse, producing nearly static videos. This arises because it evaluates only the marginal distribution $p(x_t)$ of each frame independently. Since the condition frame $x_{t-1}$  is already realistic, the generator can cheat the causal discriminator by copying $G(x_{<t}) = x_{t-1}$, resulting in little or no motion.
>
> Consistent with this analysis, the APT2-style baseline on Wan 2.1 exhibits clear motion degradation compared to our asymmetric design (Table 2). Notably, APT2 mitigates this issue by “recycling generated frames as inputs for the next autoregressive step,” introducing an explicit dependence on past frames to drive motion. Their project page also shows failure cases with limited motion when such recycled inputs are absent, further supporting our explanation.
>
> **Q2: GPU memory, GPU hours, and overhead in Stage III.**
>
> **A2:** Yes. Under the same Stage III setting (64 H20 GPUs, 8 per node, Ulysses-style context parallelism with cp=8), the bidirectional discriminator uses ~1040 GB peak GPU memory (total) and requires ~49 hours of training, while the causal discriminator baseline uses ~830 GB and ~65 hours.
>
> The higher memory cost of the bidirectional discriminator comes from processing full sequences. However, it can leverage FA3 for faster attention, whereas the causal discriminator relies on FlexAttention for causal masking, leading to longer training time.
>
> **Q3: Analysis of DMD warm-up Stage II.**
>
> **A3:** Thank you for this important question. We agree that the number of Stage II warm-up steps is sensitive, and we will include additional clarification and analysis in the final version.
>
> Due to the mode-seeking nature of reverse KL in DMD, in our I2V 1-step setting we observe that prolonged DMD training quickly leads to motion collapse, with generation degenerating to the first frame. This behavior is consistent with recent findings [1,2,3]. We therefore adopt early stopping in Stage II.
>
> Following your suggestion, we conduct a sensitivity analysis by varying the number of Stage II warm-up steps and reporting the final performance after full three-stage training in terms of image quality and dynamic degree. As shown below, increasing the warm-up length initially improves image quality, reaching a strong performance at 150 steps. Beyond this point, the dynamic degree drops sharply, indicating increasing motion collapse in downstream generation.
>
> We select 150 steps as it provides the best trade-off between image quality and motion preservation, serving as a stable initialization for Stage III.
>
> | final performance in Stage III | Image quality | dynamic degree |
> | --- | --- | --- |
> | 0 step | 62.81 | 46.81 |
> | 50 step | 64.31 | **48.29** |
> | 150 step | **70.42** | 45.35 |
> | 200 step | 70.31 | 39.29 |
> | 300 step | 69.34 | 30.42 |
>
>
> [1] Lu Y, Ren Y, Xia X, et al. Adversarial distribution matching for diffusion distillation towards efficient image and video synthesis[C]//Proceedings of the IEEE/CVF International Conference on Computer Vision. 2025: 16818-16829.
>
> [2] Zheng K, Wang Y, Ma Q, et al. Large scale diffusion distillation via score-regularized continuous-time consistency[J]. arXiv preprint arXiv:2510.08431, 2025.
>
> [3] Fan X, Qiu Z, Wu Z, et al. Phased DMD: Few-step Distribution Matching Distillation via Score Matching within Subintervals[J]. arXiv preprint arXiv:2510.27684, 2025.

---

> > ### Author Rebuttal · Reviewer_8zm4 · 2026-04-03
> >
> > Thanks for the detailed experiments.
> >
> > I suggest to add the new ablation studies to the paper.
> >
> > And I support the acceptance of this paper.

---

> > > ### Author Response · Authors · 2026-04-03
> > >
> > > Thank you very much for your supportive follow-up and for taking the time to review our additional experiments carefully.
> > >
> > > We are glad that the new ablations helped address your concerns, and we sincerely appreciate your suggestion to incorporate these results into the paper.
> > >
> > > We also truly appreciate your support for acceptance. Your feedback helped us significantly strengthen both the empirical evidence and the clarity of the paper.

---

### Official Review · Reviewer_JDL3 · 2026-03-13

**Soundness:** 3
**Presentation:** 3
**Significance:** 3
**Originality:** 2
**Overall Recommendation:** 3
**Confidence:** 5

**Summary:**

This paper presents AAD-1, asymmetric adversarial distillation for one-step image-to-video generation. The main argument is that prior distillation methods, such as APT-2 uses casual discriminators, making them poor at detecting global temporal failures. The paper proposes to keep the generator casual but replace the discriminator with a bidirectional full-video critic. The training largely follows Self-Forcing style training: 1) ODE init, 2) DMD with self roll outs, and 3) adversarial distillation. On VBench-I2V, the method shows improvements over other baselines.

**Compliance With Llm Reviewing Policy:**

Affirmed.

**Final Justification:**

The paper is solid and well-presented, and it addresses a relevant problem. That said, I'm still not fully convinced on novelty. The core contribution reads more like a careful combination of existing ideas than a genuinely new direction. The rebuttal clarified most of my concerns, and I have a better understanding of the work now, so while I'm keeping my score at 3, I wouldn't object to acceptance given the other reviewers' more positive take.

**Key Questions For Authors:**

- Will the entire AAD-1 code/models be released? I understand that it’s highly challenging to do a 1-step distillation (I highly value the author’s engineering efforts to stabilize this), but it’s also true that there are several closed-source papers/products (like the authors mentioned), APT-2, that did it. In line 146, the authors point out that APT2 relies on a closed-source model. I think authors should only mention that if they are open-sourcing AAD-1. As stabilizing adversarial training is non-trivial, open-sourcing the work will strengthen the contribution.

- Including the latency/throughput of this work will be helpful (as it’s expected to be much faster than 4-step models). Adding training details and total compute would also be helpful.

**Limitations:**

Yes, the paper adequately discusses the limitations.

**Strengths And Weaknesses:**

**Strength**
-  Achieving high-quality casual video generation with 1-step sampling is highly relevant these days. While DMD-based distillation is common for 3 or 4-step distillation, integrating adversarial optimization is recently gaining strength for fewer steps (i.e., 1-step) sampling. Distilling diffusion to work at 1-step is known to be highly challenging, and the authors have done it with non-trivial engineering.

- The core idea is well-grounded and technically sensible. Discriminator ablations clearly exhibit the importance of keeping the bidirectional discriminator of adversarial distillation.

**Weaknesses**
- While I understand 1-step distillation is a highly challenging task, this paper builds upon several existing papers with overlaps. The paper’s main claim is about the importance of a bidirectional discriminator, but I think there are prior works, including Self Forcing that uses bidirectional discriminator. Having a causal autoregressive few-step generator and a bidirectional critic for producing distribution level loss is widely considered standard (CausVid/Self Forcing’s fake critic is also modelled to be bidirectional), and only APT-2 adopts a unique design of having a causal discriminator. Other than that, I don’t find any technically new components. The discriminator design is mostly consistent with Self Forcing’s GAN variant and also shares many common components with APT-2.

- Training adversarial distillation is known to be challenging and generally requires a much bigger batch with careful hyperparameter choice. To stabilize the training the authors start with Self Forcing style initialized weights with DMD and further progresses to the adversarial stage. I feel this is a decoupling / minor enhancement from Self Forcing’s GAN variant where the generator is first initialized via DMD and then later fine-tuned with GAN loss. Adding experiment results for adversarial distillation without DMD (second stage) will be beneficial.

- Clarification on the image-to-video generation task. Self Forcing and CausVid only officially support T2V variants and I wonder how the adaptations were made. It seems like AAD-1 started from a native Wan 2.1 I2V. Did Self forcing and CausVid re-trained with I2V baselines for fairness or they simply evaluated via autoregressive roll-out style image conditioning? In T2V setups, it’s known that generating the first frame is often the most challenging part and subsequent frames can be asymmetrically inference with lower sampling steps (e.g., 4-step for the first frame and 1-step for the remainder). Further details on this would be helpful.

---

> ### Author Rebuttal · Authors · 2026-03-31
>
> We thank the reviewer for recognizing the **“non-trivial engineering”** and the **“well-grounded and technically sensible idea”** of our work. We also appreciate the constructive feedback and address the concerns below.
>
> **Q1: Will the AAD-1 code and models be released?**
>
> **A1:** Yes. Upon acceptance, we will release the AAD-1 code and models, establishing the first open-source baseline for adversarial-based single-step autoregressive video generation and facilitating future research in the community.
>
> **Q2: Latency/throughput, training details, and total compute.**
>
> **A2:** Following the reviewer’s suggestion, we report latency and throughput, evaluated on a single H100 GPU following the Self-Forcing protocol. The single-step setting significantly improves efficiency: latency is reduced by 59% (1.3B) and 59.8% (14B), while throughput increases by 145% and 150%, respectively.
>
> |  | 1.3B | 1.3B | 14B | 14B  |
> | --- | --- | --- | --- | --- |
> |  | Latency (s) | Throughput (FPS) | Latency (s) | Throughput (FPS) |
> | 1 NFE | 0.289 | 43.37 | 1.134  | 14.33 |
> | 4 NFE | 0.714  | 17.7 | 2.822 | 5.71 |
>
> In addition to the implementation details in Lines 232–258 (right column), we will include the following in the final version. The full training takes approximately 3.5 days on 64 H20 GPUs: 0.5 day for Stage I, 1 day for Stage II, and 2 days for Stage III. We use Ulysses-style context parallelism (size 8) and PyTorch checkpointing to alleviate GPU memory constraints.
>
> **W1: Discussion with Self Forcing’s GAN and APT-2.**
>
> **A1:** We agree that prior works such as Self Forcing and APT-2 explore related directions, and we will clarify the distinctions more explicitly in the final version.
>
> While Self Forcing’s GAN variant adopts a bidirectional discriminator, it is primarily validated in a few-step setting (e.g., 4-step). Extending it to the 1-step regime remains challenging in practice, as such designs often become unstable and struggle to model coherent video dynamics without additional modifications.
>
> APT-2 also considers a single-step setting, but adopts a causal discriminator. This design tends to emphasize short-range temporal consistency between adjacent frames, while providing weaker constraints on longer-range structure, which can make large motion harder to model. As described in their paper, it “recycles generated frames as inputs for the next autoregressive step,” which introduces an explicit dependency on previous generated frames to guide motion.
>
> Our contribution is not a new standalone component, but a practical recipe for **making 1-step causal video distillation work reliably**. Specifically, we combine **(1) an asymmetric bidirectional discriminator** and **(2) a phased training strategy** with a distribution-matching warm-up. We find that this combination is important for stabilizing training and achieving strong performance in the 1-step regime, **where existing designs do not directly apply.**
>
> We will revise the paper to better position our method with respect to these prior works.
>
> **W2: Ablation on warm-up Stage II.**
>
> **A2:** Per the reviewer’s suggestion, we will include the following ablation on the warm-up stage in the final version:
>
> |  | Aesthetic Quality | Image Quality |
> | --- | --- | --- |
> | w/o warm-up Stage II | 53.63 | 62.81 |
> | w/ warm-up Stage II (ours) | 59.70 | 70.42 |
>
> In the 1-step setting, we find this warm-up stage to be particularly important for stabilizing training rather than a minor enhancement. After Stage I (causal adaptation), the 1-step generator remains far from the real video distribution, often producing blurry frames and inconsistent temporal transitions. Directly applying adversarial training at this stage leads to unstable optimization, as the discriminator can easily separate real and generated samples. The DMD-based warm-up (Stage II) reduces this distribution gap, enabling stable adversarial training in the final stage.
>
> We will clarify this distinction and include the ablation to better support the role of Stage II.
>
> **W3: Clarification on I2V setups.**
>
> **A3:** Thank you for raising this point. Following CausVid’s claimed zero-shot I2V **capability** (Section 5.3, *Applications*, paragraph *Image to Video Generation*), we insert the first frame into the KV cache and then perform autoregressive rollout to generate subsequent frames. The same protocol is used for CausVid, Self Forcing, and our model to ensure a fair comparison.
>
> We agree that, in T2V settings, generating the first frame is often the most challenging step and can benefit from asymmetric inference strategies. In our I2V setup, however, the first frame is provided as a condition, so this issue does not arise, and all methods are evaluated under the same generation protocol.

---

> > ### Author Rebuttal · Reviewer_JDL3 · 2026-04-02
> >
> > Thanks for discussing a lot of my questions. My initial concerns are mostly addressed. Ablation on warm-up Stage II is also helpful, and I understand the author's claim: "Our contribution is not a new standalone component, but a practical recipe for making 1-step causal video distillation work reliably".
> >
> > While I highly acknowledge the value of practical recipes and authors' efforts to make a 1-step generation work, I am not very convinced about the originality, as the core contribution (as the authors mentioned), are (1) asymmetric bidirectional discriminator (which is already discussed in many other papers including Self Forcing), and (2) phased training strategy (which is essentially, adding GAN-based 1-step distillation on top of DMD, and does not tackle the real issue of difficulty of direct 1-step distillation). Phased training is practical, but I don't feel like the work is touching the fundamental challenges in training of GAN; it still uses 64+ GPUs, a large batch size (typical setup as shown in APT-2), and circumvents fundamental issues by starting from an already DMD initialized model.
> >
> > Overall, I feel this paper is a very borderline paper, and I will keep my initial score of 3. Nonetheless, I do agree that this phased training could be practically useful, and would not mind if it gets accepted.

---

> > > ### Author Response · Authors · 2026-04-02
> > >
> > > Thank you for your thoughtful follow-up! We’re happy to see that your initial concerns are mostly addressed, and that our ablation on warm-up Stage II is helpful! We also appreciate your acknowledgment of the value of our practical recipes. Thanks!
> > >
> > > We would like to further discuss about our contribution. Specifically, we want to highlight our work as the first systematic characterization of how discriminator backbone and supervision granularity jointly shape training behavior in 1-step causal video distillation, a setting that is qualitatively different from prior few-step or non-causal regimes.
> > >
> > > Specifically, our Stage III ablations show that:
> > > (i) under APT-2-style adversarial training without explicit recycle, purely frame-wise supervision tends to bias the model toward near-static or weak-motion videos, indicating that frame-level signals with a causal backbone is insufficient to constrain video-level dynamics.
> > > (ii) a causal discriminator backbone, with a video wise logit training signal, is less effective at detecting temporal drift and long-range inconsistencies.
> > > (iii) combining a bidirectional backbone with both frame-level and video-level logits provides complementary supervision, improving both local visual fidelity and global temporal coherence beyond the partial design choices explored in prior work.
> > >
> > > We therefore believe that the paper’s originality lies not only in introducing phased training strategy as a standalone idea, but also in providing a new empirical and mechanistic understanding of why prior discriminator design choices from Self Forcing and APT-2 fail or succeed in 1-step causal video distillation. In this sense, our contribution is not merely an engineering combination of existing ingredients, but a regime-specific design recipe supported by ablations that identify the necessary components and clarify their functional roles.
> > >
> > > Regardless of the final outcome, we sincerely thank you for your professional feedback and time, and we hope our work can help advance research on 1-step autoregressive video generation.

---

### Decision · Program_Chairs · 2026-04-30

**Decision:**

Accept (regular)

**Comment:**

The paper addresses the challenging and highly relevant problem of one-step autoregressive image-to-video generation. The reviewers highlighted several key strengths, including the well-motivated design, a sensible phased training strategy, and strong empirical results.

Following the rebuttal, most reviewer concerns were resolved, resulting in largely positive assessments.

Reviewer JDL3 maintained a reservation regarding the originality of the contribution. While acknowledging the engineering effort and the practical usefulness of the proposed phased training recipe, the reviewer argued that the core components are primarily combinations of existing techniques and do not address the fundamental challenges of GAN training. The authors responded by clarifying that their work provides a necessary, practical recipe to successfully extend multi-step methods to a single-step regime. They also provided analysis demonstrating how their approach differs from APT-2, specifically regarding the "recycling" of generated frames.

Reviewer u6aq noted a minor concern regarding the generalizability of the conclusions about the discriminator architecture. The authors committed to clarifying the scope of their claims in the final version, better emphasizing that the findings are specific to the 1-step causal video distillation setting.

AC Recommendation:
In view of the above, the AC thinks that the concerns regarding originality alone are not sufficient grounds for rejection, given the method's demonstrated effectiveness in an understudied topic. The paper's merits outweigh its drawbacks, and the AC recommends acceptance.

Recommendations for the Camera-Ready Version:

The authors are encouraged to:

* Include APT-2 in the experimental comparisons where it fits, or explicitly document the rationale for its exclusion, given its high relevance to this work.
* Follow through on their explicit commitment to release all code and models upon acceptance to support reproducibility.